# Realized Heritability, Risk Assessment, and Inheritance Pattern in *Earias vittella* (Lepidoptera: Noctuidae) Resistant to Dipel (*Bacillus thuringiensis* Kurstaki)

**DOI:** 10.3390/toxins14100686

**Published:** 2022-10-06

**Authors:** Syed Faisal Ahmad, Asim Gulzar, Naeem Abbas, Muhammad Tariq, Intazar Ali, Abdulwahab M. Hafez

**Affiliations:** 1Department of Entomology, PMAS-Arid Agriculture University, Rawalpindi 44000, Pakistan; 2Pesticides and Environmental Toxicology Laboratory, Department of Plant Protection, College of Food and Agriculture Sciences, King Saud University, Riyadh 11451, Saudi Arabia; 3Department of Entomology, Faculty of Agriculture and Environment, The Islamia University of Bahawalpur, Baghdad Ul Jadeed Campus, Bahawalpur 63100, Pakistan

**Keywords:** laboratory selection, Dipel resistance, dominance expression, polygenic response, spotted bollworm

## Abstract

*Earias vittella* Fabricius is a potential cotton and okra pest in South Asia. The realized heritability, risk assessment, and inheritance mode of *Bacillus thuringiensis* Kurstaki (Btk) resistance were determined in the Dipel-selected (DIPEL-SEL) *E. vittella*. The DIPEL-SEL strain had a 127.56-fold rise in Dipel resistance after nine generations compared to the laboratory reference strain (LAB-PK). The overlapping of 95% fiducial limits in the median lethal concentrations (LC_50_s) of the F_1_ (DIPEL-SEL♂ × LAB-PK♀) and F_1_^ǂ^ (DIPEL-SEL♀ × LAB-PK♂) suggested a lack of sex linkage and an autosomal Dipel resistance. The dominance (D_LC_) values for the F_1_ (0.86) and F_1_^ǂ^ (0.94) indicated incompletely dominant resistance to Dipel. Backcrossing of the F_1_♀ × Lab-PK♂ revealed a polygenic response of resistance to Dipel. The realized heritability estimation (*h*^2^) of resistance to Dipel was 0.19. With 20% to 90% selection mortality, the generations required for a tenfold increase in LC_50_ of Dipel were 4.7–22.8, 3.1–14.9, and 2.3–11.1 at *h*^2^ of 0.19, 0.29, and 0.39, respectively, and a constant slope of 1.56. At slope values of 2.56 and 3.56 with a constant *h*^2^ = 0.19, 7.7–37.4 and 10.6–52.0 generations were needed to increase the tenfold LC_50_ of Dipel in the DIPEL-SEL *E. vittella*. It is concluded that the DIPEL-SEL *E. vittella* has an autosomal, incompletely dominant, and polygenic nature of resistance. The *h*^2^ of 0.19 suggested that a high proportion of phenotypic variation for the Dipel resistance in *E. vittella* was heritable genetic variation. The present results will support the creation of an effective and suitable resistance management plan for better control of *E. vittella*.

## 1. Introduction

*Earias vittella* Fabricius (Lepidoptera: Noctuidae), commonly known as spotted bollworm, is a major cotton and okra pest in South Asia [1,2]. Additionally, *E. vittella* has other hosts such as *Althaea rosea* (Malvaceae), *Abutilon indicum* (Malvaceae), *Hibiscus rosasinensis* (Malvaceae), and *Malva parviflora* (Malvaceae) [3]. This pest remains active year round and completes six to eight generations [2]. The maximum infestation of this pest occurs in the months of August and September [2,3]. *E. vittella* larvae feeds on buds, squares, shoots, flowers, and green cotton bolls, and bores down into the terminal buds of shoots in okra, which ultimately results in the wilting and stunting of the plant. It causes approximately 40% and 41.6% losses in cotton seed and okra, respectively [4,5], while heavy infestations cause up to 50% losses in cotton yield [2].

Various synthetic insecticides are sprayed to control *E. vittella* on okra and cotton [2,4,6]. Because of the inappropriate usage of synthetic insecticides, this pest has developed resistance to pyrethroids, organophosphates, and new chemistry insecticides [2,7,8]. *Bacillus thuringiensis* insecticides have long been suggested as alternatives to synthetic insecticides for the management of pests including *E. vittella*, because they are ecofriendly, economical, and target-specific. *Bacillus thuringiensis kurstaki* cry proteins have played a vital role in protecting cotton from bollworm attacks [9,10]. In Pakistan, cultivation of transgenic cotton with single gene (i.e., Cry1Ac) began during 2002 [11], but it was adopted at a large scale in 2010 [12]. The use of *B. thuringiensis* against cotton bollworms has become the most promising alternative control to synthetic insecticides [13]. The cry toxins produced by *B. thuringiensis* provide an effective way to control lepidopteran pests through genetically modified crops and have no or minimal harmful effects to human beings, natural fauna, and other nontargeted organisms [14]. However, extensive adoption and continued use of single-trait products may contribute toward resistance evolution among different insect species. *B. thuringiensis* field-evolved and laboratory-selected resistance has been reported in *Spodoptera frugiperda* (Smith) [15], *Helicoverpa armigera* (Hübner) [10,16], *Pectinophora gossypiella* (Saunders) [17,18], *Helicoverpa zea* (Boddie) [19], *Plutella xylostella* (Linnaeus) [20], and *Trichoplusia ni* (Hübner) [21].

Realized heritability (*h*^2^) is the ratio between additive genetic variance (V_A_) and phenotypic variance (V_P_). Lower *h*^2^ values reveal high phenotypic variation and less additive genetic variation [22,23,24]. Higher phenotypic variance may come from gene mutations and laboratory selection pressure, but in field environments it may come from migration of individuals, insecticide alternation, and ecological factors [23,25,26]. The *h*^2^ estimation is a useful variable in proactive and effective resistance management of insect pests [27,28]. The *h*^2^ estimation of resistance through laboratory selection predicts the rate of phenotypic and genetic variations due to insecticide resistance [28,29]. By using laboratory selection results and *h*^2^ values, the assessment of risk of resistance to chemicals is also important for proactively to define insecticide resistance management (IRM) strategies [28,30,31]. Previously, various studies have described these parameters in different insect pests [28,29,32,33].

Knowledge about the *B. thuringiensis* resistance inheritance is important in devising resistance management strategies and their future implementation to delay resistance [18,20,34]. Inheritance analyses including autosomal or sex-linked genes, frequency of resistant alleles, and dominance expression levels explore the nature of developed resistance and are crucial to success the current *B. thuringiensis* resistance management strategies [18,27]. There are numerous reports on the inheritance mode of *B. thuringiensis* toxins resistance in pests such as *P. xylostella* [20,35], *H. armigera* [36,37], *P. gossypiella* [38,39], *O. furnacalis* [40], and *Helicoverpa zea* [41].

Dipel, a commercial formulation of *B. thuringiensis*, produces a blend of protoxins effective against a wide range of caterpillars and bollworms. It contains crystalline *B. thuringiensis* insecticidal proteins and spores that become activated when exposed to the alkaline medium of an insect’s gut upon ingestion by lepidopteran larvae [42]. Recently, a low-to-high level of field-evolved Dipel resistance (6 to 111-fold) was also described in *E. vittella* [43]. Characterization of Dipel resistance is useful in building successful IRM plans to prolong the efficacy of Dipel against *E. vittella*. The aims of this study were (1) to explore the risk of Dipel resistance by laboratory selections, (2) to quantity the *h*^2^ values, and (3) to find the inheritance mode (autosomal, dominance pattern, and frequency of resistant alleles) of Dipel resistance in *E. vittella*.

## 2. Results

### 2.1. Dipel Resistance Selection

The LC_50s_ of Dipel for the LAB-PK, Field-POP, and DIPEL-SEL were 1.32-, 41.70-, and 168.38-µg/mL, respectively. The toxicity of Dipel was significantly different in Field-POP compared with the LAB-PK (non-overlapped 95% FL). A Field-POP of *E. vittella* revealed 31.59-fold resistance ratio to Dipel compared to the LAB-PK. In addition, selection of *E. vittella* with Dipel increased 127.56-fold resistance in the DIPEL-SEL (G9) compared to the LAB-PK (Table 1).

### 2.2. Realized Heritability (h^2^) Estimation 

The estimated *h*^2^ of resistance to Dipel was 0.19 in the DIPEL-SEL (G_9_) *E. vittella* strain (Table 2).

### 2.3. Dipel Resistance Projected Rate

With 20% to 90% selection mortality and constant slope of 1.56, the generations (G) required for a tenfold increase in LC_50_ of Dipel were 4.7–22.8, 3.1–14.9, and 2.3–11.1 at *h*^2^ of 0.19, 0.29, and 0.39, respectively (Figure 1). At slopes of 2.56 and 3.56 with a constant *h*^2^ = 0.19, 7.7–37.4 and 10.6–52.0 generations were needed to increase the tenfold LC_50_ of Dipel in the DIPEL-SEL *E. vittella* (Figure 2). These results suggest that the risk of developing resistance to Dipel may become high if the selection pressure is continued intensively. 

### 2.4. Sex Linkage and Degree of Dominance of Dipel Resistance

The LC_50s_ of Dipel for the F_1_, F_1_^ǂ^, BC_1_, and BC_2_ were 85.75-, 127.76-, 57.34-, and 35.87-µg/mL, respectively. The RRs were 64.96-, 96.78-, 43.43-, and 27.17-fold for the F_1_, F_1_^ǂ^, BC_1_, and BC_2_, respectively, compared with the LAB-PK. Overlapping of 95% FLs in the LC_50_s of the F_1_ and F_1_^ǂ^ revealed an autosomal Dipel resistance and lack of sex linkage. D_LC_ values for the F_1_ (0.86), F_1_^ǂ^ (0.94), BC_1_ (0.78), and BC_2_ (0.68) directed incompletely dominant nature of resistance to Dipel (Table 3).

### 2.5. Effective Dominance (D_ML_) of Dipel Resistance in E. vittella

Effective dominance (D_ML_) values revealed that resistance to Dipel at 1024 µg/mL was completely recessive (D_ML_ = 0.00) and incompletely recessive at 512 µg/mL (D_ML_ = 0.22). However, Dipel resistance was expressed as incompletely dominant at lower doses of 256 µg/mL, 128 µg/mL, 64 µg/mL, and 32 µg/mL (D_ML_ = 0.65, 0.75, 0.77, and 0.81, respectively (Table 4).

### 2.6. Number of Genes Involved in Dipel Resistance

Result of the backcross (BC_2_) denoted significant differences between the observed and expected larval mortality in five concentrations out of six concentrations (*p* ≤ 0.05). This suggested polygenic nature of Dipel resistance in the DIPEL-SEL *E. vittella* (Table 5).

## 3. Discussion

Recently, low-to-high levels of Dipel resistance have been found in different *E. vittella* populations [43]. In the present study, a 127.56-fold resistance to Dipel was detected in a DIPEL-SEL *E. vittella* after eight generations of selection in comparison to the LAB-PK. This rise of Dipel resistance in *E. vittella* proposes the significant effects on progression of resistance by selection pressure. The likely reason for the rise in Dipel resistance is the existence of resistant alleles in the field population [43]. *B. thuringiensis* crystal protein (*Bt*. formulated insecticide or *Bt.* trait plant) resistance has been reported in numerous insect pests, including *P. xylostella* [44], *S. frugiperda* [45], *H. armigera* [10], *D. virgifera* [46], *P. gossypiella* [47], *H. zea* [19], and *O. nubilalis* [48].

Realized heritability (*h*^2^) estimation provides an indication towards the development of resistance to chemicals in the selected individuals of insect pests under laboratory conditions [27,28]. In the present study, the estimated *h*^2^ value of 0.19 suggests that a high proportion of phenotypic variation was not due to genetic differences, and that *E. vittella* has a lower tendency to develop Dipel resistance genetically. The low *h*^2^ value reveals that a larger number of *E. vittella* generations is needed to develop substantial resistance against Dipel under field conditions. This result also shows that Dipel is probably still effective against *E. vittella*, and the rate of resistance development could be lower in the field, if applied according to recommendations. Our results are in agreement with the other findings such as *D. virgifera* resistant to Cry3Bb1 (*h*^2^ = 0.16) [49] and Cry1Ac- and *Btk*-resistant *H. armigera* (*h*^2^ = 0.10 and 0.23, respectively) [50]. However, contrary to present results, higher *h*^2^ values have been described in Cry1Ac- and Cry1Ab-resistant *P. xylostella* (*h*^2^ = 0.80 and 0.92, respectively) [51], Cry1Ab-resistant *Chilo suppressalis* (Walker) (*h*^2^ = 0.52) [52], and Cry1Ac-resistant *Chrysodeixis includens* (Walker) (*h*^2^ = 0.72) [53]. Assessment of chemical resistance risk by G = 1/*h*^2^S is an imperative finding to establish rational resistance management strategies [29,31]. The risk of developing resistance to *Bt*. toxins has been explored previously in some insects [28,54]. The present result reveals that G of 4.7–22.8, 3.1–14.9, and 2.3–11.1 are needed to create tenfold increase in LC_50_ of Dipel at *h*^2^ of 0.19, 0.29, and 0.39, respectively, by 20–90% selection intensity (mortality) and a constant slope of 1.56. The G of 7.7–37.4 and 10.6–52.0 equate to slopes of 2.56, and 3.56, at a constant *h*^2^ = 0.19. These findings suggest that the risk of Dipel resistance increases with an increase in *h*^2^ value. Therefore, prudence is necessary when considering the measures to minimize Dipel resistance and to control *E. vittella*.

The progression of resistance to insecticides can be influenced by the degree of dominance of resistant genes. For example, the insecticide resistance controlled by the recessive or incompletely recessive genes grows slowly, while the insecticide resistance controlled by dominant or incompletely dominant genes develops rapidly [27,55,56]. The current study showed that the Dipel resistance is expressed as an autosomal and incompletely dominant feature in *E. vittella*. An autosomal inheritance of *B. thuringiensis* crystal protein resistance has been reported in many insect pests [35,36,38,39,57]. The degree of dominance can fluctuate on the basis of type of *B. thuringiensis* crystal proteins, genetic backgrounds, individual species, ecological environments, and diverse selection histories [58]. For instance, incompletely or completely recessive resistance to Cry1Ac in *H. armigera* [59,60], *P. gossypiella* [18,38], Cry1Ab and Cry1F in *O. nubilalis* [61,62], and Cry1F, Cry2Ab2, and Cry1A in *S. frugiperda* [63,64,65] but incompletely or completely dominant resistance to Cry1Ac [10,37] and Cry2Ab [36] in *H. armigera*, and Cry1Ab in *Mythimna unipuncta* (Haworth) [66] and *B. fusca* [67] have been reported. Our results showed incompletely dominant resistance; therefore, *B. thuringiensis* insecticides with different modes of action should be rotated to reduce the rapid increase in Dipel resistance. For the D_ML_, it was observed that as Dipel concentration increased from 32 to 1024 µg/mL, dominance level shifted from incompletely dominant to completely recessive. Normally, with the increase in concentrations of Dipel, the D_ML_ decreased in the present results. For example, resistance to Dipel was incompletely dominant at concentrations of 32-, 64-, 128-, and 256-µg/mL, incompletely recessive at a concentration of 512-µg/mL and completely recessive at a concentration of 1024-µg/mL. These data suggest that dominance level varies with the change in concentrations of any insecticide. Moreover, the D_ML_ levels observed in this study directs that higher doses of Dipel could make resistance incompletely and completely recessive, and so can be employed to minimize the resistance occurrence in *E. vittella*. The concentration-dependent dominance was also observed in other studies [34,68,69].

Insecticide resistance may develop as a monogenic or polygenic trait in insect pest populations [56,70]. Nevertheless, the occurrence of a polygenic type of insecticide resistance is more expected in the laboratory-selected pest populations, because the screening pressure within the population phenotypes favors a polygenic response [68,71]. Normally, the polygenic type of resistance grows slowly and dilutes quickly by the breeding of susceptible and resistant individuals in comparison with the monogenic type of resistance [31,72]. This dilution of resistant alleles by hybridization boosts the flow of susceptible alleles [58,73]. In this study, the backcrosses revealed polygenic nature of Dipel resistance in *E. vittella*. Previously, a polygenic nature of resistance inheritance was reported for Cry2Ab in *H. armigera* [36], Cry1Ab, Cry1F and Cry1Ah in *Ostrinia furnacalis* (Guenée) [40,74,75], Cry1Ab in *M. unipuncta* [66], and Btk in *T. ni* [21]. Contrary to our results, a monogenic nature of resistance to *B. thuringiensis* insecticides has also been described in different pests [10,39,57,64,65]. The polygenic resistance observed in this study is advantageous because the multiple-gene-driven resistance is easier to overwhelm compared to single-gene-driven resistance.

In conclusion, the selection of *E. vittella* with Dipel led to high resistance that is expressed as an autosomal, incompletely dominant, and polygenic trait. The incompletely dominant resistance suggests that *E. vittella* has the possibility to build high Dipel resistance, and therefore should be rotated with different action-mode insecticides to slow the development of resistance. Moreover, the data of D_ML_ suggest that the use of higher doses of Dipel can make resistance incompletely or completely recessive in *E. vittella*, which is easy to control. Multiple-gene-driven resistance is also advantageous over single-gene-driven resistance, as this type of resistance is easier to overcome. The lower realized heritability value (*h*^2^ = 0.19) of resistance to Dipel showed higher phenotypic variations and a genetically slower chance of resistance development in the field. Additionally, nonchemical control measures should also be adopted for the better management of *E. vittella*. In future, the enzymatic investigations (cytochrome P450 monooxygenases, esterases, and glutathione S transferases) and the possible genetic mechanisms responsible for developing resistance in insects to Bt.-based products should be expanded to better characterize the resistance of *E. vittella*.

## 4. Materials and Methods

### 4.1. E. vittella Collection and Rearing Protocol

An *E. vittella* population was collected in 2016 from a field of okra with no *B. thuringiensis* application in Rawalpindi, Punjab, Pakistan (33.723961° N, 72.904578° E). Then, this population was reared for 25 generations in the laboratory unexposed to any insecticide and used as the susceptible reference strain (LAB-PK). Another population of *E. vittella*, designated as Field-POP, was sampled from the infested cotton (non-Bt variety) fields in Multan, Punjab, Pakistan (30.1575° N, 71.5249° E) during 2018–2019. The susceptibility of LAB-PK (G_25_) and Field-POP (G_1_) to Dipel was measured and found to be significantly different, as reported in Ahmad et al. [43]. Furthermore, the Field-POP was continuously selected with Dipel for eight generations in the laboratory to make a resistant line (named as DIPEL-SEL). About 250 full-grown *E. vittella* larvae were collected and reared on okra pods in a laboratory at 25 ± 2 °C temperature, 60 ± 5% RH, and photoperiod of 14/10 h (L/D) as previously described in Ahmad et al. [43]. Larvae were maintained in plastic boxes (15 cm × 25 cm) covered with a muslin cloth to prevent larvae escape. Small pieces of okra pods (3–4 cm) were washed with water, dried at room temperature, and then provided to larvae ad libitum until pupae formation. Newly formed pupae were shifted to other unused plastic boxes and tightened with a muslin cloth to stop the escape of adults. After emergence, adults were shifted to plastic jars (18 cm × 35 cm) and fed on ten percent sugar solution. The strips of nappy liner were hung into the jars for egg deposition and mouths of jars were tightly covered with a muslin cloth. Nappy liner strips were harvested every day and placed in plastic boxes for hatchability. Fresh okra pods were provided to neonates. Second-instar stage of larvae was subjected for bioassays. Completely randomized design (CRD) was used for the experiments in the laboratory.

### 4.2. Insecticides

The commercial formulation of *Bacillus thuringiensis kurstaki* (Dipel^®^ 54DF, Valent Biosciences, Libertyville, IL, USA) was used for bioassays and selection of *E. vittella.*

### 4.3. Selection with Dipel

Before starting selection experiment, a preliminary bioassay was conducted on *E. vittella* field population (Field-POP) at G_1_ to estimate the level of resistance and desired concentration values required for starting selection. In each selection, 2nd-instar larvae were exposed to different concentrations (Appendix A) of Dipel for eight generations (G_1_ to G_8_) using diet immersion method [43]. Okra pods were immersed into the solution of Dipel concentrations for 10 s and aerated at room temperature for 30 min. Averages of 600 larvae were screened in each generation of selection. Mortality data were recorded after 5 days and survivors were continued to obtain next progeny.

### 4.4. Genetic Crosses

For determination of inheritance mode of Dipel resistance, reciprocal crosses and backcrosses were made between the DIPEL-SEL and LAB-PK strains. Sexes were separated on pupal stage and were identified by presence of well-developed knob-like outgrowth at the anterodorsally end of male cocoon [76]. Within 24 h adult emergence, two reciprocal crosses—F_1_ (30 DIPEL-SEL♂ × 30 LAB-PK♀) and F_1_^ǂ^ (30 DIPEL-SEL♀ × 30 LAB-PK♂)—were made for determining the degree of dominance (D_LC_). Two backcrosses were made with the parents—BC_1_ (30 F_1_♀ × 30 LAB-PK♂) and BC_2_ (30 F_1_♀ × 30 DIPEL-SEL♂)—to know the frequency of resistant alleles involved in Dipel resistance. All strains were reared at aforementioned laboratory environments.

### 4.5. Bioassays

The toxicity of Dipel was determined on newly molted 2nd-instar *E. vittella* larvae using diet immersion bioassay, previously described in Ahmad et al. [43]. Six to eight serial concentrations of Dipel with five replicates were made in distilled water. Fresh okra pieces (3 to 4 cm in length) were dipped for 10 s in each tested solution of insecticide and air-dried for half an hour. Treated okra pieces were putted in Petri dishes having filter papers on the surface (5 pieces/Petri dish). Ten larvae to a replicate, 50 larvae to a concentration, and 300–400 larvae to a bioassay were exposed. For control, okra pods were immersed in distilled water only. Mortality was noted after five days of exposure [51].

### 4.6. Degree of Dominance (D_LC_) and Effective Dominance (D_ML_)

D_LC_ for Dipel resistance was measured as described in Bourguet et al. [77].
(1)$$D_{\mathrm{LC}}=\frac{\left(\log{\mathrm{LC}}_{50F1}-{\mathrm{logLC}}_{50\mathrm{LAB}-\mathrm{PK}}\right)}{\left({\mathrm{logLC}}_{50\mathrm{DIPEL}\;-\mathrm{SEL}}-{\mathrm{logLC}}_{50\mathrm{LAB}-\mathrm{PK}}\right)}$$

The effective dominance (D_ML_) was calculated using the following formula Bourguet et al. [77]:D_ML_ = MT_F1_ − MT_LAB−PK_/MT_DIPEL−SEL_ − MT_LAB−PK_(2)

MT is the percent mortality on a single Dipel dose for a given population. D_LC_ or D_ML_ ranges from 0 to 1: D_LC_ or D_ML_ value of zero illustrates completely recessive resistance, D_LC_ or D_ML_ value of 1 illustrates completely dominant resistance, D_LC_ or D_ML_ value of ≤0.5 or >0 illustrates incompletely recessive resistance, and D_LC_ or D_ML_ value of <1 or >0.5 illustrates incompletely dominant resistance.

### 4.7. Number of Resistant Genes Involved in Dipel Resistance

Chi-square (χ^2^) analysis of backcross (BC_2_) was performed to test the null hypothesis of monogenic response of Dipel resistance [78] as follows:(3)$$\chi^{2}=\frac{{\left(F-\mathrm{pn}\right)}^{2}}{\mathrm{pqn}}$$
where ‘F’ indicates observed mortality of BC_2_ to Dipel concentrations, n indicates the number of larvae treated to Dipel concentrations, p indicates the expected mortality, and q was estimated as 1 − p. The statistically significant difference (*p ≤* 0.05) between observed and expected mortality against more than 50% tested concentrations showed polygenic nature of Dipel resistance.

### 4.8. Realized Heritability (h^2^)

‘*h*^2^’ was estimated as described in Tabashnik [28] as:(4)$$h^{2}\;=R/S$$

‘R’ denotes selection response and was estimated as: (5)$$R=\frac{{\mathrm{Log}\;\mathrm{LC}}_{50\mathrm{DIPEL}-\mathrm{SEL}\;\left(G9\right)}-{\mathrm{Log}\;\mathrm{LC}}_{50\;\mathrm{Field}-\mathrm{Pop}\;\left(G1\right)}}{n}$$
where ‘n’ shows generations screened with Dipel.

‘S’ denotes selection differential and was calculated as:(6)$$S=i\times \sigma_{p}$$
where ‘*i*’ denotes selection intensity estimated from the percent survival of selection (*p*) using selection intensity table reported by Falconer [22]. ‘*σ_p_*’ denotes phenotypic deviation and determined as:(7)$$\sigma_{p}=\frac{1}{\mathrm{mean}\;\mathrm{slope}\;\left(G1-G9\right)}$$

Generations (G) required for tenfold increase in LC_50_ of Dipel were estimated as: (8)$$G\;=\frac{1}{h^{2}S}$$

Projected rate of Dipel resistance was estimated between selection intensity and G at calculated and assumed *h*^2^ and slope.

### 4.9. Data Analyses

Toxicity data were analyzed by R software [79]. The formula of Abbott [80] was used to correct the control mortality, if had. A log-logistic function was used to determine the concentration response for *E. vittella* populations [81]. Natural logs of Dipel concentrations were taken to fulfill the theory of normal distribution of residuals. A function “dose *p*” from the package “MASS” was used to calculate the lethal concentrations (LC_50_) and 95% confidence intervals (CIs) [82]. LC_50_ values were found to be significant if their CIs at 95% did not overlap [83]. Resistance ratios (RR) were determined by the formula:RR = LC_50_ of Dipel for the DIPEL − SEL/LC_50_ of Dipel for the LAB-PK(9)

## Figures and Tables

**Figure 1 toxins-14-00686-f001:**
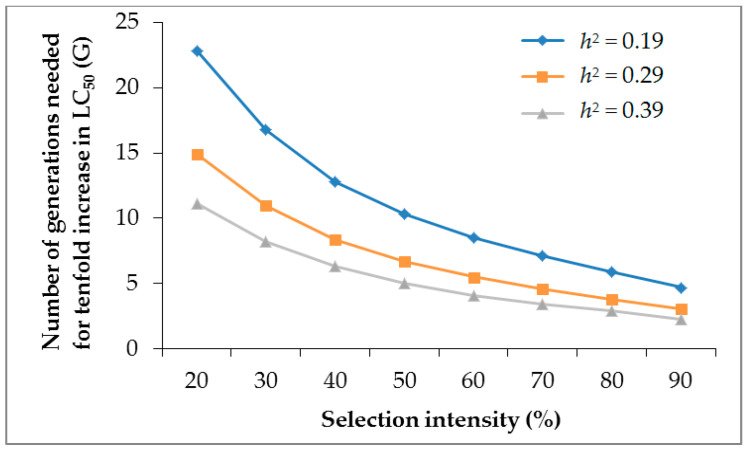
Effect of mortality (selection intensity) on Dipel resistance in the DIPEL-SEL strain of *Earias vittella* at different heritability values.

**Figure 2 toxins-14-00686-f002:**
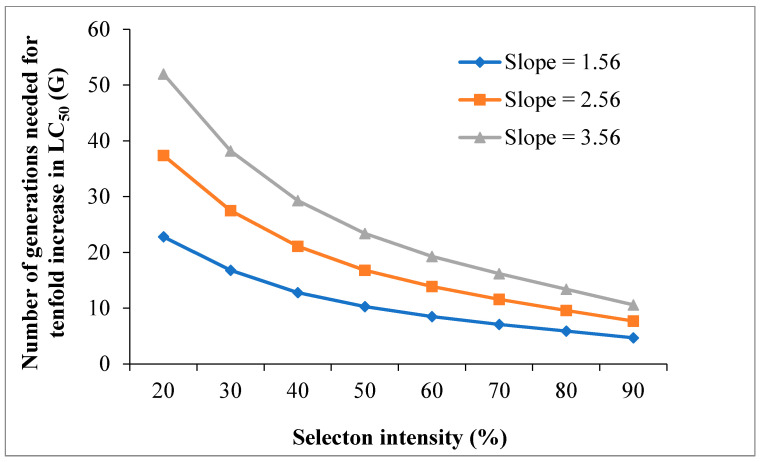
Effect of mortality (selection intensity) on Dipel resistance in the DIPEL-SEL strain of *Earias vittella* at different slope values.

**Table 1 toxins-14-00686-t001:** Response of the LAB-PK, Field-POP, and DIPEL-SEL *E. vittella* to Dipel.

Population	LC_50_ (95% FL) (µg/mL)	Slope ± SE	N	RR
LAB-PK (G_25_) *	1.32 (1.10–1.59)	1.71 ± 0.18	300	1.00
Field-POP (G_1_) *	41.70 (34.11–50.98)	1.45 ± 0.14	400	31.59
DIPEL-SEL (G_9_)	168.38 (139.77–202.85)	1.67 ± 0.17	400	127.56

LC_50_ = median lethal concentration, FL = fiducial limits; N = number of larvae exposed in bioassay; RR = resistance ratio, LC_50_ of Dipel for Field-POP or DIPEL-SEL/LC_50_ of Dipel for LAB-PK; * published data by Ahmad et al. [43].

**Table 2 toxins-14-00686-t002:** Realized heritability (*h*^2^) of Dipel resistance in *Earias vittella*.

Generation	Insecticide	* Initial LC_50_	* Final LC_50_	Selection Response (*R*)	Percent Survival (*p*)	Selection Intensity (*i*)	Initial Slope	Final Slope	Mean Slope	Phenotypic Deviation (*σ**_p_*)	Selection Differential (*S*)	*h* ^2^
9 (G_1_–G_9_)	Dipel	1.62	2.23	0.07	65.29	0.56	1.45	1.67	1.56	0.64	0.36	0.19

* Initial and final LC_50_ (μg/mL) are the Log LC_50_ of the field population (G_1_) and DIPEL-SEL (G_9_), respectively.

**Table 3 toxins-14-00686-t003:** Dominance of Dipel resistance in *E. vittella*.

Population	Insecticide	LC_50_ (95% FL) (µg/mL)	Slope ± SE	N	RR	D_LC_
F_1_ (DIPEL-SEL♂ × LAB-PK♀)	Dipel	85.75 (69.99–105.07)	1.40 ± 0.14	400	64.96	0.86
F_1_^ǂ^ (DIPEL-SEL♀ × LAB-PK♂)	Dipel	127.76 (101.85–160.25)	1.20 ± 0.12	400	96.78	0.94
BC_1_ (F_1_♀ × DIPEL-SEL♂)	Dipel	57.34 (46.25–71.10)	1.30 ± 0.13	400	43.43	0.78
BC_2_ (F_1_♀ × LAB-PK♂)	Dipel	35.87 (29.67–43.36)	1.58 ± 0.15	400	27.17	0.68

FL = fiducial limits, N = number of larvae exposed in each bioassay including control; RR = resistance ratio, LC_50_ of Dipel in the F_1_, F_1_^ǂ^, or BC/LC_50_ of Dipel in the LAB-PK; D_LC_ = degree of dominance.

**Table 4 toxins-14-00686-t004:** Effective dominance (D_ML_) of Dipel resistance in *E. vittella*.

Concentration (µg/mL)	Strain	Mortality (%)	D_ML_
1024	LAB-PK	100	0.00 Completely recessive
	DIPEL-SEL	96	
	F_1_ (R♂ × S♀)	100	
512	LAB-PK	100	0.22 Incompletely recessive
	DIPEL-SEL	82	
	F_1_ (R♂ × S♀)	96	
256	LAB-PK	100	0.65 Incompletely dominant
	DIPEL-SEL	66	
	F_1_ (R♂ × S♀)	78	
128	LAB-PK	100	0.75 Incompletely dominant
	DIPEL-SEL	41	
	F_1_ (R♂ × S♀)	56	
64	LAB-PK	100	0.77 Incompletely dominant
	DIPEL-SEL	22	
	F_1_ (R♂ × S♀)	40	
32	LAB-PK	100	0.81 Incompletely dominant
	DIPEL-SEL	6	
	F_1_ (R♂ × S♀)	24	

**Table 5 toxins-14-00686-t005:** Model of Dipel resistance inheritance using chi-square analysis in BC_2_
*Earias vittella*.

Concentration (μg/mL)	Number of Larvae	Observed Mortality (Proportion)	* Expected Mortality (Proportion)	χ^2^ [df = 1]	*p*
16	50	11 (0.22)	2.50 (0.05)	0.19	0.139
32	50	20 (0.40)	7.50 (0.15)	7.91	0.005
64	50	28 (0.56)	15.50 (0.31)	20.87	<0.001
128	50	36 (0.72)	26.00 (0.52)	51.21	<0.001
256	50	43 (0.86)	36.00 (0.72	122.50	<0.001
512	50	48 (0.96)	44.50 (0.89)	387.28	<0.001
				∑591.96 (df = 5)	<0.001

* Expected response = 1/2 (number of dead F_1_ larvae + number of dead DIPEL-SEL larvae). Mortalities differed significantly at *p* ≤ 0.05.

## Data Availability

Not applicable.

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
