# Peer review of "Realized Heritability, Risk Assessment, and Inheritance Pattern in Earias vittella (Lepidoptera: Noctuidae) Resistant to Dipel (Bacillus thuringiensis Kurstaki)"

_toxins, 2022, doi:10.3390/toxins14100686_

Round 1

Reviewer 1 Report

In this paper, the authors analyzed the realized heritability, risk assessment and inheritance mode of Bt resistance in Earias vittella under the indoor selection. They found that DIPEL-SEL E. vittella is high resistant to Dipel with an autosomal, incompletely dominant, and polygenic trait. However, there are some writing errors that need to be corrected and improved.

1.     Page 2, paragraph 1: The first two sentences refer to synthetic pesticides, but the third sentence is about Bt. There is no connection or transition between the sentences.

2.     Page 5, Line 110: h2 should be “h2”. The same errors appear in Figure 1.

3.     Figure 1: LC50, 50 should be written with subscript; The abscissa does not correspond to the points in the line; The figure should provide a more detailed description.

4.     Table 2: σp, the “p” should be written with subscript. The same errors appear in Equation 7.

5.     Page 7,Equation 2:wrong writing. Since there is no subscript, the whole formula cannot be understood.

Author Response

Reviewer 1

Comments and Suggestions for Authors

In this paper, the authors analyzed the realized heritability, risk assessment and inheritance mode of Bt resistance in Earias vittella under the indoor selection. They found that DIPEL-SEL E. vittella is high resistant to Dipel with an autosomal, incompletely dominant, and polygenic trait. However, there are some writing errors that need to be corrected and improved.

Response: Thanks for the comments; we carefully read the manuscript for writing errors.

  1. Page 2, paragraph 1: The first two sentences refer to synthetic pesticides, but the third sentence is about Bt. There is no connection or transition between the sentences.

Response: Thanks for the suggestion, made connection

  1. Page 5, Line 110: h2 should be “h2”. The same errors appear in Figure 1.

Response: Corrected as suggested

  1. Figure 1: LC50, 50 should be written with subscript; The abscissa does not correspond to the points in the line; The figure should provide a more detailed description.

Response: Done as suggested

  1. Table 2: σp, the “p” should be written with subscript. The same errors appear in Equation 7.

Response: Done as suggested

  1. Page 7,Equation 2:wrong writing. Since there is no subscript, the whole formula cannot be understood.

Response: Done subscripts

Reviewer 2 Report

The authors reported realized heritability, risk assessment,and inheritance pattern in Earias vittella(Lepidoptera: Noctuidae) resistant to Dipel (Bacillus thuringiensis kurstaki). The results may provide suggestions for the resistance management in E. vittella. However, factually this study is not a novel finding to the present knowledge, and similar study on other lepidopteron was reported more than 20yrs ago. In addition, there are several questions in the manuscript that may be affect the results reported here, but the authors do not design the experiment to eliminate the factors. I think the present manuscript could not meet the standard of the journal.

Major question:

1. The selected population was not derived from the susceptible strain. The differences in the background of the resistant and susceptible strains may affect the results in this manuscript. The authors may design experiment to eliminate the differences.

2. Although the information on the population was mentioned in the manuscript, in material and methods section the clear population information can not be easily obtained by readers.

Minor questions:

3. Table 5: in the legend of the table reported monogenic model of Dipel resistance inheritance using chi-square analysis in BC2 Earias vittella. But in the manuscript, they said the inheritance is controlled by polygenic nature.

4. L138:(DML= 0.65,0.85,77, and 81 respectively (Table 4). There are some errors in the reported data compared with the Table 4.

5. There are many errors on the words and expression. The authors should carefully check the manuscript to meet the standard of the journal. 

Author Response

Reviewer 2

Comments and Suggestions for Authors

The authors reported realized heritability, risk assessment, and inheritance pattern in Earias vittella (Lepidoptera: Noctuidae) resistant to Dipel (Bacillus thuringiensis kurstaki). The results may provide suggestions for the resistance management in E. vittella. However, factually this study is not a novel finding to the present knowledge, and similar study on other lepidopteron was reported more than 20yrs ago. In addition, there are several questions in the manuscript that may be affect the results reported here, but the authors do not design the experiment to eliminate the factors. I think the present manuscript could not meet the standard of the journal.

Major question:

  1. The selected population was not derived from the susceptible strain. The differences in the background of the resistant and susceptible strains may affect the results in this manuscript. The authors may design experiment to eliminate the differences.

Response: We thank Reviewer#2 and agree with him/her about the importance of elimination any doubt regards the susceptibility of our LAB-PK strain. And that why we: first, had derived LAB-PK strain from a field population that was collected from a free Bt cotton field (no Bt application and the cotton variety was non-Bt). Second, we reared this field population without any kind of insecticide exposure for 25 generations to end up with the LAB-PK strain. Third, we carefully reviewed and followed the key literatures in this area who applied this approach (these literatures were published in well reputed international journals and we provided below some examples). Fortunately, the different results of LAB-PK (G25) compare to the results of Field-POP (G1) and DIPEL-SEL (G9) have supported our assumption about its susceptibility.

Examples of relevant literatures:

  1. Shah et al. 2015, Acta Tropica 142: 149–155
  2. Shad et al. 2010, Pest Manag Sci. 66(8):839-46.
  3. Sayyed et al. 2005, Pest Manag Sci., 61:636–642
  4. Wang et al. 2009, Pest Manag Sci., 65: 629–634
  5. Achalekea and Brevault 2010, Pest Manag Sci., 66: 137–141
  6. Although the information on the population was mentioned in the manuscript, in material and methods section the clear population information cannot be easily obtained by readers.

Response: Thanks for the comment, population information cleared in material and method section.

Minor questions:

  1. Table 5: in the legend of the table reported monogenic model of Dipel resistance inheritance using chi-square analysis in BC2 Earias vittella. But in the manuscript, they said the inheritance is controlled by polygenic nature.

Response: Thanks for the comment, we used monogenic model described in many articles, in this model if the null hypothesis is rejected, it revealed the polygenic resistance. This was clearly described in the section 4.7. Number of resistant genes involved in Dipel resistance.

  1. L138 (DML= 0.65,0.85,77, and 81 respectively (Table 4). There are some errors in the reported data compared with the Table 4.

Response: Thanks for the comment, corrected

  1. There are many errors on the words and expression. The authors should carefully check the manuscript to meet the standard of the journal.

Response: Thanks for the comments; we carefully read the manuscript for writing errors.

Reviewer 3 Report

The current manuscript focuses on the spotted bollworm resistance attributes to Dipel. The authors studied realized heritability, risk assessment and inheritance patterns of the insect to Dipel exposure under lab conditions using different insect populations. The authors found that DIPEL-SEL E. vittella has an autosomal, incompletely dominant, and polygenic nature of resistance. Also, the results suggest that a high proportion of phenotypic variation was not due to genetic differences and E. vittella has lower tendency to develop Dipel resistance genetically. The study is very well designed and conducted. Materials and methods are super clear. The results were very clearly interpreted. It is one of the few manuscripts which keep the reader’s interest high and up. The only fact missing in the discussion is about the possible genetic mechanisms responsible for developing resistance in insects to Bt based products.

Author Response

Reviewer 3

Comments and Suggestions for Authors

The current manuscript focuses on the spotted bollworm resistance attributes to Dipel. The authors studied realized heritability, risk assessment and inheritance patterns of the insect to Dipel exposure under lab conditions using different insect populations. The authors found that DIPEL-SEL E. vittella has an autosomal, incompletely dominant, and polygenic nature of resistance. Also, the results suggest that a high proportion of phenotypic variation was not due to genetic differences and E. vittella has lower tendency to develop Dipel resistance genetically. The study is very well designed and conducted. Materials and methods are super clear. The results were very clearly interpreted. It is one of the few manuscripts which keep the reader’s interest high and up. The only fact missing in the discussion is about the possible genetic mechanisms responsible for developing resistance in insects to Bt based products.

Response: Thanks for the good comments, the possible genetic mechanisms responsible for developing resistance in insects to Bt. based products will be explored in future studies. We mentioned in the discussion.